# Analyzing the Association between *Candida* Prevalence, Species Specificity, and Oral Squamous Cell Carcinoma: A Systematic Review and Meta-Analysis— Candida and OSCC

Shankargouda Patil

Department of Maxillofacial Surgery and Diagnostic Sciences, Division of Oral Pathology, College of Dentistry, Jazan University, Jazan 45142, Saudi Arabia; dr.ravipatil@gmail.com

**Abstract:** The present review is a qualitative and quantitative analysis of the overall prevalence of *Candida*, and its species specificity in oral squamous cell carcinoma (OSCC). PubMed, Scopus, and Web of Science databases were searched using the keywords '*Candida* and oral squamous cell carcinoma'. Only case-control observational studies in the English language evaluating the prevalence and species specificity of *Candida* in OSCC were included. 297 articles were identified (PubMed-106, Scopus-148, Web of Science-43) using the keywords. After screening the titles and abstracts, 206 articles were removed as they were duplicates (118) or irrelevant to the topic (88). Full text of the remaining 91 articles was assessed using the inclusion criteria, based on which only seven articles were included in the systematic review. For the quantitative analysis, the odds ratio and confidence interval were assessed and a forest plot was generated. Based on the meta-analysis, there is an association between the total *Candida*, *Candida albicans* (*CA*) and OSCC, while the association with non-*Candida albicans (NCA)* is relatively weak. The number of studies included in the meta-analysis was relatively low (four to five). Further, at least one of the studies included in the meta-analysis for the association of *CA.*, *NCA* and total *Candida* with OSCC had a Newcastle–Ottawa score below 7. Thus, although the results showed an association, the quality and quantity of the evidence may not be sufficient for conclusive inference.

**Keywords:** *Candida*; *Candida albicans*; oral squamous cell carcinoma

## 1. Introduction

Oral squamous cell carcinoma (OSCC) is a multifactorial disease, with tobacco and alcohol being the most common independent risk factors [1–5]. Apart from these known risk factors, several factors (microbiome; lifestyle factors including obesity, diet, and occupation; environmental factors like air pollution, heavy metal exposure, and genetic susceptibility) have been implicated in the development of OSCC [6–29], although their evidence of association is limited. Thus, such factors are termed potential risk factors. One such factor for OSCC is microorganisms. Studies have shown the presence of several oral microbes in OSCC patients [6–20]. Unlike external factors like tobacco and alcohol, the presence of oral microbes in an OSCC case could be a result of the microbe being a commensal or a secondary infection in the cancerous tissue. Thus, eliciting the role of microorganisms in OSCC would first require the establishment of a significantly higher association of the microbe in OSCC compared to healthy patients with normal oral mucosa. If a significant association is noted in OSCC, then the microbe in question can be assessed for the presence of carcinogenic properties. Among oral microbes, much importance has been given to human papillomavirus (HPV) types 16 and 18, due to their proven association with cervical and oropharyngeal cancer [30–32]. On the contrary, the role of

HPV in OSCC remains inconclusive due to discrepancies in the results of individual studies [6]. Similar to HPV, microbes implicated in cancer in other anatomical locations including Helicobacter pylori in gastric cancer, have been assessed for its association with OSCC [33]. In most cases, the studies have been inconclusive, which is largely attributed to the differences in the sensitivity and specificity of the diagnostic modalities used to detect the microbes. In addition, if the oral microbe in question is present in a higher proportion as a commensal in the control (healthy individuals with normal oral mucosa) population, the results of the comparison between OSCC and the control often turn to insignificant or in few studies, even an inverse correlation has been recorded [20]. Due to these limitations, analyzing the association between oral microbes and OSCC has been largely inconclusive. One of the most commonly studied oral microbes in OSCC is *Candida* [16–19,34–40]. *Candida albicans* have shown to produce carcinogens including nitrosamines, which can activate proto-oncogenes triggering carcinomatous changes [41]. *Candida* is also capable of breaking down ethanol in to acetaldehyde, which in turn can cause adducts in protein and DNA [42–45]. These DNA adducts have shown to interfere with DNA replication resulting in point mutations, and aberrations of the chromosome [46]. Acetaldehyde has also shown to affect the cytosine methylation and DNA repair enzyme causing cell cycle aberrations which may culminate in tumor development [47]. In addition, acetaldehyde can increase the reactive oxygen species by binding with anti-oxidative peptide glutathione increasing the risk of DNA damage. The mitochondrial damages induced by the acetaldehyde have also shown to increase the reactive oxygen species, and key survival factors including NF-kb, which can in turn increase the risk of tumour development [48,49]. The acetaldehyde level considered to be carcinogenic is 4100 μM [43]. Among the *Candida* aspecies, *Candida albicans* (CA) and a few non-*Candida albicans* (NCA) *including Candida parapsilosis* and *Candida tropicalis* have shown to exceed this threshold level [41]. Despite multiple observational studies being published on the prevalence and species specificity of *Candida* in OSCC, the association of *Candida* with OSCC remains to be inconclusive, which again is due to the same limitations as suggested for *H. pylori* and HPV based studies. In addition, *Candida* pathogenesis and treatment sensitivity also vary significantly based on the species [50]. Thus, establishing a mere association between *Candida* and OSCC would not be sufficient. It would be necessary to identify both the Candida prevalence and species specificity in OSCC. Unlike *Candida albicans (CA)*, which is regarded as a commensal, *non-Candida albicans (NCA)* such as *C. tropicalis, C. glabrata, C. dubliniensis, C. krusei,* and *C. parapsiolosis* have shown to exhibit inherent treatment resistance and are implicated in deteriorating the oral health [18,19,50]. Thus, *NCA* is designated as pathogenic microbes and their predominance in the oral cavity is considered as a sign of oral dysbiosis. A shift in the oral *Candida* flora towards NCA in OSCC could provide preliminary evidence of a potential pathogenic association between *Candida* and OSCC. Thus, the present systematic review assessed the published literature for studies investigating the total and species-specific *Candida* prevalence in OSCC. Based on the data obtained, the *Candida* prevalence and species specificity in OSCC were analyzed both qualitatively and quantitatively. The rationale for the meta-analysis was to estimate using the current published data if there is a significant difference in the *Candida* prevalence and species specificity between oral cancer tissue and the normal oral mucosa. A significantly higher *Candida* prevalence, and or a predominant *NCA* in OSCC compared to normal oral mucosa, would provide a preliminary evidence that *Candida*, especially the treatment-resistant *NCA* have an affinity towards OSCC, although it would not be possible to comment if the higher prevalence and/or *NCA* specificity is a result of the OSCC or the OSCC is a result of the higher prevalence and/or *NCA* specificity. The null hypothesis of the review was that there was no association between *Candida* and OSCC.

## 2. Materials and Methods

The systematic review strictly adhered to the PRISMA (Preferred Reporting Items for Systematic Review and Meta-Analysis) protocol.

### 2.1. Inclusion Criteria

Observational case-control studies in the English language. Studies using individuals with clinically assessed normal oral mucosa (NOM) as the control group, and histopathologically confirmed cases of OSCC as the study group.

### 2.2. Exclusion Criteria

Studies other than case-control observation studies such as interventional studies. Reviews, individual case reports/series, abstract from conferences, commentaries, letters. Studies in a language other than English. Studies analyzing microorganisms other than *Candida*, and diseases other than OSCC.

The inclusion and exclusion criteria were framed r\to enable the identification of only the observational studies comparing the prevalence of *Candida* between OSCC and the normal oral mucosa using various diagnostic modalities.

### 2.3. Focused Question

Is there an association between *Candida* prevalence, species specificity, and OSCC?

The framework of the population (P), intervention (I), comparison (C), outcome (O), studies (S) were used for this focused question. P represents histopathologically confirmed cases of OSCC; I represents the diagnostic modality used to assess the presence of *Candida*; C represents clinically determined cases of NOM; O represents the *Candida* prevalence, species specificity in OSCC compared to the control group (NOM); S represents case-control observational studies.

### 2.4. Search Strategy

The SCOPUS, PubMed, and Web of Science databases were searched using the keywords "*Candida* and oral squamous cell carcinoma" until October 2019.

### 2.5. Studies Selection and Data Extraction

The review was conducted independently by two reviewers (SGP and ATR). The review consisted of two steps:

Step 1: The titles and abstracts of all identified articles were screened and the duplicates and irrelevant articles were excluded.

Step 2: The full text of the screened articles was assessed using the inclusion criteria.

Only those articles fulfilling the inclusion criteria and providing sufficient details to conduct the qualitative analysis were included.

### 2.6. Risk of Biased Assessment

The quality of the included studies was assessed using the Newcastle–Ottawa scale (NOS). Parameters including the comparability and selection outcome and exposure were considered to score the articles. The selection criteria had a maximum score of 4, the comparability criteria had a maximum score of 2, and the maximum score for the outcome, exposure criteria was 4. In total, a maximum of score 10 could be achieved by an article. Studies with scores higher than 7 were considered good.

### 2.7. Statistical Analysis

The Kappa coefficient was estimated to evaluate the inter-observer bias between the two reviewers (SGP and ATR). In addition to the systematic review, the data extracted was also subjected to meta-analysis. For the quantitative analysis, the confidence interval (CI) and odds ratio (OR) of the included studies were calculated. Only those studies with compatible data were subjected to quantitative analysis.

## 3. Results

### 3.1. Study Selection

Keywords '*Candida* and oral squamous cell carcinoma', identified 297 (PubMed-106, Scopus-148, Web of Science-43) articles.

Step 1: Screening the titles and abstracts of the identified article revealed 118 duplicates and 88 irrelevant articles.

Step 2: The remaining 91 articles were subjected to full-text assessment using the selection criteria.

Only seven studies [34–40] satisfied the inclusion criteria and provided sufficient data for qualitative analysis. Figure 1 provides a summary of the search strategy employed in the systematic review. Values of 0.98 and 1 were found as the inter-reviewer Kappa coefficient values for the first and second steps of the review respectively. Data extracted from the included studies are summarized in Table 1.

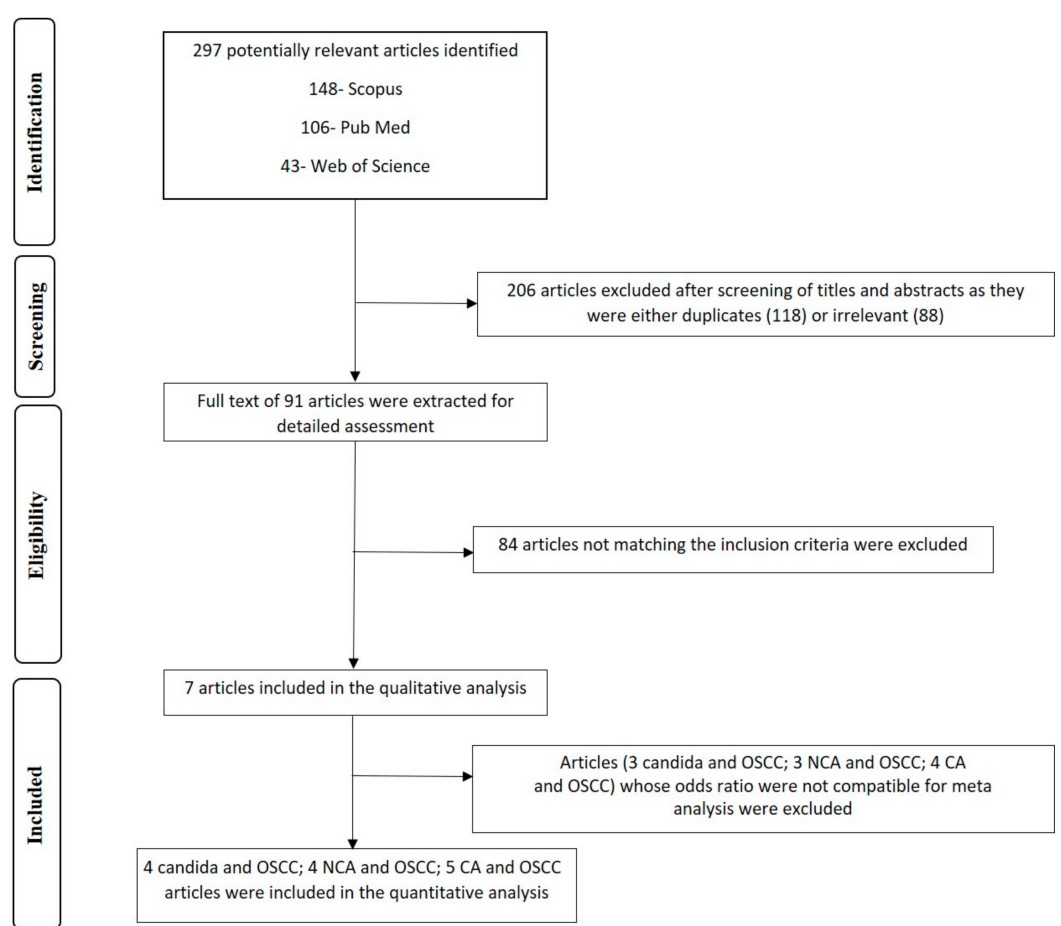

**Figure 1.** Summary of the search strategy employed.

**Table 1.** Summary of the data extracted from the studies included in the systematic review.

| First Authors Name/Year/Country [Reference Number] | Comparison Groups | | | Comparison Group Matching | Sample Collected | Modalities Used | Results of *Candida*, *CA*, and *NCA* |
|---|---|---|---|---|---|---|---|
| Hulimane/2018/India [34] | OSCC-18 | OED-32 | NOM-50 | Details not provided | Oral swab | SDA, CHROMagar | *Candida* positive in OSCC-18, OED-32, NOM-2 <br> *CA* positive in OSCC-10, OED-7, NOM-2 <br> *NCA* positive in OSCC-8, OED-25, NOM-0 |
| | **Additional information:** *C. tropicalis* present in OSCC-6, OED-14, NOM-0; *C. glabrata* present in OSCC-2, OED-10, NOM-0. <br> Mixed colonies: *CA* and *C. glabrada* present in OSCC-0, OED-2, NOM-0. | | | | | | |
| Roy/2019/India [35] | OSCC-40 | OPMD-30 | NOM-25 | Gender matched. Age was not matched | Oral swab | SDA, CHROMagar | *Candida* positive in OSCC-31, OPMD-12, NOM-5 <br> *CA* positive in OSCC-4, OPMD-7, NOM-4 <br> *NCA* positive in OSCC-18, OPMD-5, NOM-1 |
| | **Additional information:** *C. krusei* present in OSCC-10, OPMD-5, NOM-1; *C. glabrata* present in OSCC-6, OED-10, NOM-0; *C. topicalis* present in OSCC-2, OPMD-0, NOM-0. <br> Mixed colonies: *C. glabrata* and *C. krusei* present in OSCC- 1, OPMD-0, NOM-0; *C. tropicalis* and *C. krusei* present in OSCC-3, OPMD-0, NOM-0; *C. tropicalis* and *C. glabrata* present in OSCC-1, OPMD-0, NOM-0; *C. albicans* and C. tropicalis present in OSCC-3, OPMD-0, NOM-0;*C. krusei*, *C. glabrata* with *C. albicans* present in OSCC-1, OPMD-0, NOM-0. | | | | | | |
| Alnuaimi/2015/Australia [36] | OSCC-52 | | NOM-104 | Age, gender, and denture status matched | Oral rinse | CHROMagar, Real-Time PCR | *Candida* positive in OSCC-39, NOM-51 <br> *CA* positive in OSCC-31, NOM-32 <br> *NCA* positive in OSCC-8, NOM-19 |
| | **Additional information:** *CA* genotype A present in OSCC-27, NOM-15; *CA* genotype B present in OSCC-4, NOM-16; CA genotype C present in OSCC-0, NOM-1. <br> *C. dubliniensis* present in OSCC-3, NOM-3; *C. glabrata* present in OSCC-2, NOM-3; *C. guilliermondii* present in OSCC-2, NOM-0. <br> *C. krusei* present in OSCC-1, NOM-0; *C. parapsilosis* present in OSCC-0, NOM-9; *C. tropicalis* present in OSCC-0, NOM-3; *C. lusitaniae* present in OSCC-0, NOM-1. <br> Mixed colonies: *C. albicans* genotype A and *C.glabrata* present in two cases; *C. albicans* genotype A and *C. parapsilosis* present in two cases; C. *albicans* genotype B and *C. tropicalis* present in one case; *C. albicans* genotype A and *C. lusitaniae* present in one case; *C. dubliniensis* and *C. parapsilosis* present in one case. Overall OSCC-3, NOM—seven cases had mixed colonies. The article did not mention which mixed colonies were present in which comparative group. | | | | | | |

**Table 1.** *Cont.*

| First Authors Name/Year/Country [Reference Number] | Comparison Groups | | | | Comparison Group Matching | Sample Collected | Modalities Used | Results of *Candida*, *CA*, and *NCA* |
|---|---|---|---|---|---|---|---|---|
| Sankari/2019/India [37] | OSCC-90 | | NOM-170 | | Age and gender matched | Saliva | SDA, CHROMagar, germ tube tests, chlamydospore formation on cornmeal agar, sugar assimilation, and fermentation tests | *Candida* positive in OSCC-63, NOM-34 *CA* positive in OSCC-26, NOM-19 *NCA* positive in OSCC-26, NOM-15 Mixed *CA* and *NCA* in OSCC-11, NOM-0 |
| | **Additional information:** Mixed colonies (*CA* and *NCA*) present in OSCC-11, NOM-0. | | | | | | | |
| Castillo/2018/Argentina [38] | OSCC-25 | Atypical OLP-11 | Chronic candidiasis-25 | NOM-15 | Age and gender matched | Oral swab | CHROMagar, colony morphology, sugar fermentation tests, germ tube test, morphology in maize agar, 42 °C growth | *Candida* positive in OSCC-25, atypical OLP-11, chronic candidiasis-25, NOM-15 *CA* positive in OSCC-16, atypical OLP-6, chronic candidiasis-11, NOM-15 *NCA* positive in OSCC-9, atypical OLP-2, chronic candidiasis-7, NOM-0 |
| | **Additional information:** *C. dubliniensis* present in OSCC-2, NOM-0; *C. glabrata* present in OSCC-1, NOM-0; *C. krusei* present in OSCC-4, NOM-0; *C. tropicalis* present in OSCC-1, atypical lichen planus-2, chronic candidiasis-7, NOM-0. Mixed colonies (*CA and C.tropicalis*) present in OSCC-0, atypical lichen planus-0, chronic candidiasis-2, NOM-0. | | | | | | | |
| Gupta/2019/ India [39] | OSCC-30 | OL-30 | | NOM-20 | Details not provided | Saliva | SDA, germ tube test, chlamydospore production in milk serum and in cornmeal broth +5% milk media | *Candida* positive in OSCC-14, OL-11, NOM-0*CA* positive in OSCC-11, OL-4, NOM-0 *NCA* positive in OSCC-3, OL-7, NOM-0 |
| | **Additional information:** The *NCA* identified was *C. tropicalis*. Details on mixed colonies were not provided. | | | | | | | |
| Makinen/2018/Finland [40] | OSCC-100 | | NOM-75 | | Age and gender-matched | Saliva | CHROMagar API ID 32C, and Bichro-Dubli Fumouze latex agglutination tests | *Candida* positive in OSCC-74, NOM-47 *CA* positive in OSCC-63, NOM-45 *NCA* positive in OSCC-17, NOM-11 |
| | **Additional information:** *C. dubliniensis* present in OSCC-6, NOM-4; C. tropicalis present in OSCC-3, NOM-0; *C. glabrata* present in OSCC-2, NOM-4; *C. parapsilosis* present in OSCC-2, NOM-1; *Candida sake* present in OSCC-2, NOM-0; *C. krusei* present in OSCC-1, NOM-0; *C. guillermondii* present in OSCC-1, NOM-1; *C. kefyr* present in OSCC-0, NOM-1. Mixed colonies (2 species) present in seven cases. Mixed colonies (three species) present in one case. The article did not mention which mixed colonies were present in which comparative group. | | | | | | | |

Note: *CA*—*Candida albicans*; *NCA*—*non-Candida albicans*; OSCC—oral squamous cell carcinoma; NOM—normal oral mucosa; OL—oral leukoplakia; OLP—oral lichen planus; SDA—Sabouraud dextrose agar; OPMD—oral potentially malignant disorder.

### 3.2. Study Characteristics

Four studies [34,35,37,39] were from India, one [36] from Australia, one [38] from Argentina, and one [40] from Finland. The samples collected included oral rinse, saliva, and oral swab. The modalities used to identify the *Candida* prevalence and species specificity included Sabouraud dextrose agar (SDA), CHROMagar, germ tube test, API ID 32C, and Bichro-Dubli Fumouze latex agglutination tests, chlamydospore production in milk serum and in cornmeal broth +5% milk media/corn meal agar, colony morphology, sugar assimilation, fermentation tests, morphology in maize agar, growth at 42 °C, and real-time PCR (Polymerase chain reaction).

### 3.3. Newcastle–Ottawa Scale

Definition of the cases and control were provided by all seven [34–40] included studies. Four studies [36–38,40] matched the comparative groups for known confounding factors including age and gender. One study [35] matched for gender but not for the age. The two remaining studies [34,39] either did not provide any matching details or did not match the comparative groups. Matching for potential confounding factors was done in only one study [36] wherein the comparative groups were matched for the denture status. Three [34,35,39] of the seven included studies scored less than 7. The scores given to each of the included studies are summarized in Table 2.

### 3.4. Candida Prevalence in OSCC Compared to the NOM

The odds ratio and 95% confidence interval for the total *Candida* prevalence as estimated from each of the included studies are provided in Table 3. Based on the individual odds ratio, the overall ratio was estimated for total *Candida* prevalence in OSCC as 4.75 (95% confidence interval of 1.85–12.19) as shown in Table 4. The estimated overall ratio was used to generate the forest plots for *Candida* prevalence as shown in Figure 2. The data revealed a significant association between *Candida* and OSCC.

**Table 2.** Quality of the studies included in the systematic review as assessed by the Newcastle–Ottawa Scale.

| The First Author [Reference Number] | Selection | | | | Comparability | | | Exposure | | | Total Score |
|---|---|---|---|---|---|---|---|---|---|---|---|
| | Case Definition | Case Representativeness | Control Selection | Control Definition | Matching Known Confounding Factor | Matching Potential Confounding Factor | Secure Patient Records | Interviewer Blinded to Cases and Control | The Similarity in the Case and Control Ascertainment | Non-Response Rate | |
| Hulimane [34] | 1 | 1 | 1 | 1 | 0 | 0 | 1 | 0 | 1 | 0 | 6 |
| Roy [35] | 1 | 1 | 1 | 1 | 0 | 0 | 1 | 0 | 1 | 0 | 6 |
| Alnuaimi [36] | 1 | 1 | 1 | 1 | 1 | 1 | 1 | 0 | 1 | 0 | 7 |
| Sankari [37] | 1 | 1 | 1 | 1 | 1 | 0 | 1 | 0 | 1 | 0 | 7 |
| Castillo [38] | 1 | 1 | 1 | 1 | 1 | 0 | 1 | 0 | 1 | 0 | 7 |
| Gupta [39] | 1 | 1 | 1 | 1 | 0 | 0 | 1 | 0 | 1 | 0 | 6 |
| Makinen [40] | 1 | 1 | 1 | 1 | 1 | 0 | 1 | 0 | 1 | 0 | 7 |

**Table 3.** Odds ratio and confidence interval calculated for *Candida* prevalence and species specificity from each study included in the systematic review.

| Author/Year/Country [Reference Number] | Odds Ratio | 95% Confidence Interval |
|---|---|---|
| *Candida* **and OSCC** | | |
| Roy/2019/India [35] | 13.78 | 3.54–58.27 |
| Alnuaimi/2015/Australia [36] | 3.12 | 1.42–7.09 |
| Sankari/2019/India [37] | 9.33 | 4.99–17.54 |
| Makinen/2015/Finland [40] | 1.70 | 0.84–3.41 |
| **CA and OSCC** | | |
| Hulimane/2018/India [34] | 30.00 | 5.67–158.80 |
| Roy/2019/India [35] | 0.58 | 0.10–3.51 |
| Alnuaimi/2015/Australia [36] | 3.32 | 1.57–7.05 |
| Sankari/2019/India [37] | 3.23 | 1.59–6.62 |
| Makinen/2015/Finland [40] | 1.14 | 0.59–2.20 |
| **CA Genotype A** | | |
| Alnuaimi/2015/Australia [36] | 6.41 | 2.77–14.96 |
| **CA Genotype B** | | |
| Alnuaimi/2015/Australia [36] | 0.46 | 0.11–1.54 |
| *NCA* **and OSCC** | | |
| Roy/2019/India [35] | 19.64 | 2.57–851.39 |
| Alnuaimi/2015/Australia [36] | 0.78 | 0.27–2.04 |
| Sankari/2019/India [37] | 3.66 | 1.73–7.93 |
| Makinen/2015/Finland [40] | 1.19 | 0.49–3.02 |
| *C. dubliniensis* **and OSCC** | | |
| Alnuaimi/2015/Australia [36] | 2.06 | 0.26–15.88 |
| Makinen/2015/Finland [40] | 1.13 | 0.26–5.67 |
| *C. glabrata* **and OSCC** | | |
| Alnuaimi/2015/Australia [36] | 1.35 | 0.11–12.13 |
| Makinen/2015/Finland [40] | 0.36 | 0.03–2.62 |
| *C. krusei* **and OSCC** | | |
| Roy/2019/India [35] | 8.00 | 0.99–361.35 |
| **C. parapsilosis and OSCC** | | |
| Makinen/2015/Finland [40] | 1.51 | 0.08–90.28 |
| *C. guillermondii* **and OSCC** | | |
| Makinen/2015/Finland [40] | 0.75 | 0.01–59.46 |

**Table 4.** Random effect model-based assessment of overall ratio and 95% confidence intervals for *Candida* prevalence and species specificity from the studies included in the meta-analysis.

| Overall Ratio | 95% Confidence Interval |
|---|---|
| *Candida* **and OSCC** | |
| 4.75 | 1.85–12.19 |
| **CA and OSCC** | |
| 2.75 | 1.13–6.71 |
| *NCA* **and OSCC** | |
| 1.95 | 0.73–5.20 |

Note: The overall ratio was calculated only for those parameters which had meta-analysis compatible data from at least four studies.

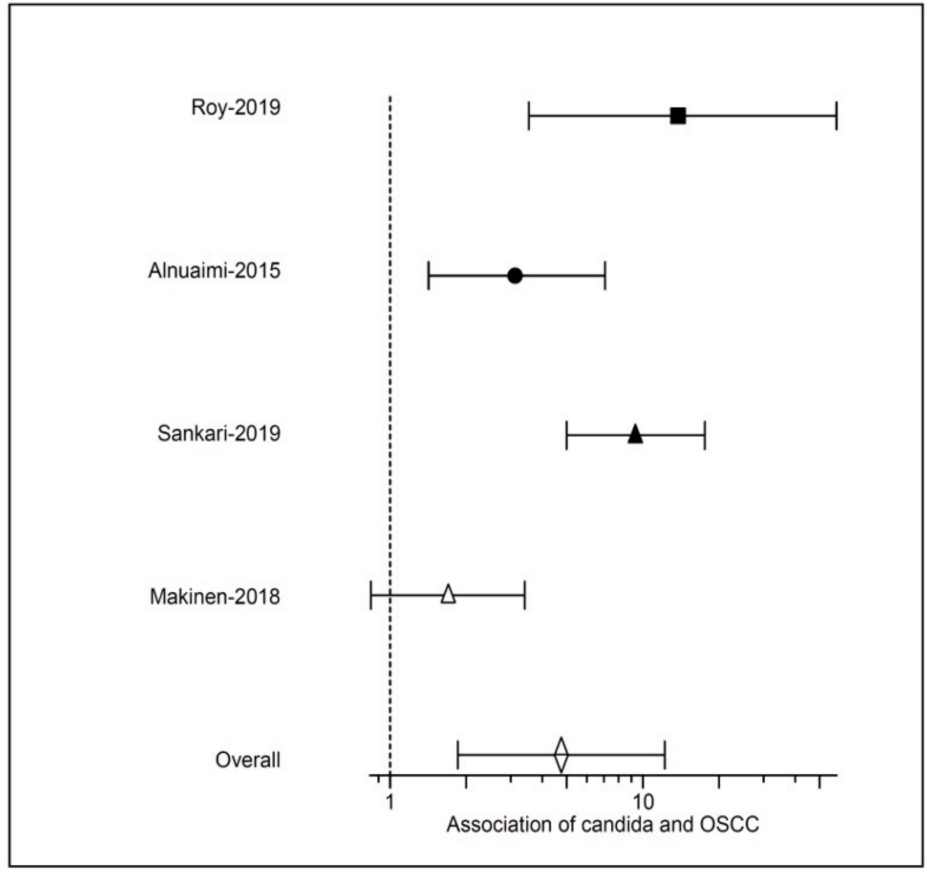

**Figure 2.** Forest plot summarizing the results of the meta-analysis for *Candida* and OSCC.

### 3.5. Candida Species Specificity in OSCC Compared to the NOM

The odds ratio and 95% confidence interval for the *Candida* species specificity as estimated from each of the included studies is provided in Table 3. Based on the individual odds ratio, the overall ratio was estimated for *CA* and *NCA* prevalence in OSCC as 2.75 (95% confidence interval of 1.13–6.71) and 1.95 (95% confidence interval of 0.73–5.20) respectively as shown in Table 4. The overall ratio of each species of the *NCA* was not calculated as the number of studies providing compatible data was limited (only one or two studies). Similar to *Candida* prevalence, the estimated overall ratio for *CA* and *NCA* was used to generate the forest plots as shown in Figures 3 and 4 respectively. The data revealed a significant association between *CA* and OSCC, while the association between *NCA* and OSCC was not significant.

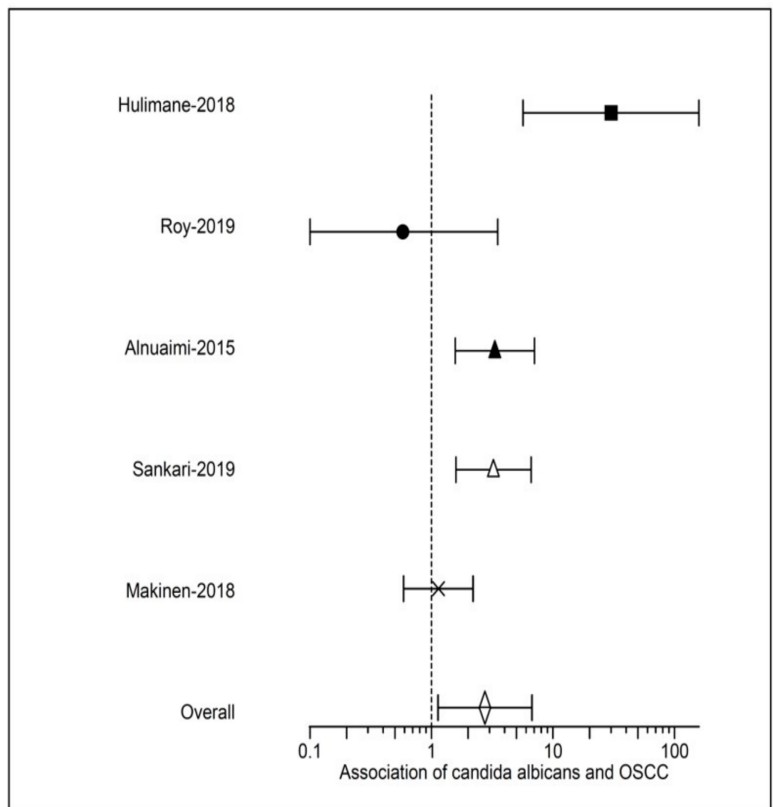

**Figure 3.** Forest plot summarizing the results of the meta-analysis for CA and OSCC.

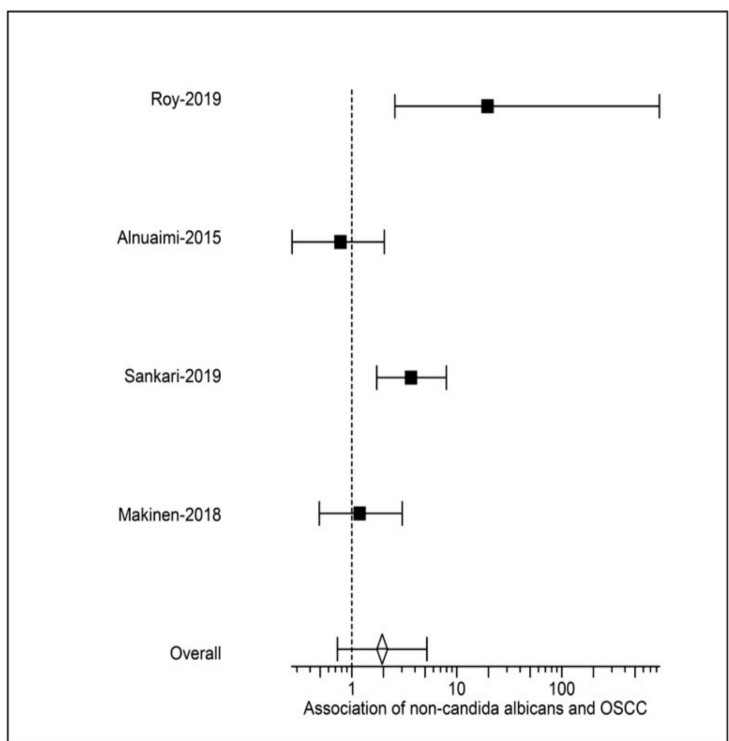

**Figure 4.** Forest plot summarizing the results of the meta-analysis for *NCA* and OSCC.

## 4. Discussion

Under physiological conditions, *CA* is considered as one of the most common *Candida* species in the oral cavity [51]. Thus, most studies on *Candida* have largely focused on *CA*. In a majority of these

*CA*-based studies, the experimental model includes the isolation and characterization of the *Candida* from the oral cavity *through* salivary sampling, oral rinse, or an oral swab [34–40]. Estimating the prevalence of *Candida* and the species specificity for different population groups is a major research strategy employed in oral *Candida*-based studies. Despite being a commensal, *CA* is a well-established opportunistic infection, capable of inducing both local and systemic infections [51]. Thus, research on the prevalence and species specificity of *Candida*, including *CA*, has been extended to a wide range of disease entities, especially with immunodeficiency states such as HIV/AIDS, and transplant patients wherein the opportunistic nature of *Candida* have a higher risk of establishing an infection [52–55].

Although common anti-fungal agents have shown to effective against *CA*, their efficiency is relatively poor against the *NCA* due to their inherent treatment resistance and virulence properties [56,57]. A shift in the *Candida* flora from *CA to NCA* has been suggested as a sign of oral dysbiosis which could potentially increase the risk of local and systemic diseases, including cancer [18,19]. Several studies have evaluated the carcinogenic potential of *Candida* through epidemiological, in vitro, and in vivo studies [34–40,58–64]. Despite evidence of producing carcinogenic agents such as acetaldehyde [64], the evidence for a causal association between *Candida* and OSCC has been largely inconclusive. The lack of evidence is in turn attributed to the varying epidemiological data from individual studies [34–40]. To overcome the limitation of individual study differences, the present literature review was formulated to provide a comprehensive estimate of *Candida* prevalence and species specificity in OSCC through qualitative and quantitative analysis.

The present systematic review included seven case-control observational studies [34–40] wherein the control were individuals with normal oral mucosa and the cases were histopathologically confirmed OSCC. Although each study had other variable groups ranging from smokers, oral leukoplakia (OL), oral potentially malignant disorder (OPMD), oral lichen planus (OLP), the review focused primarily on the results of the *Candida* prevalence and species specificity in OSCC and the healthy controls. As mentioned earlier, the most common sampling techniques were oral swab, oral rinse, and saliva. Other than oral swab, the other sampling techniques could potentially include microbial colonies from different parts of the oral cavity and may not be a true representation of the lesional area. The most common diagnostic modality used was CHROMagar. Being a relatively economical modality compared to genotyping, the CHROMagar-based chromogenic differentiation is often the most desired system of *Candida* species differentiation. The subjectivity of the observers in determining the color of the colonies in CHROMagar often leads to inter-observer bias, which is its major limitation. Thus, many studies employ additional tests to confirm the findings of the CHROMagar. Some of the additional diagnostic modalities used in the studies included in the present review are the germ tube test, API ID 32C, Bichro-Dubli Fumouze -latex agglutination tests, chlamydospore production in milk serum and in cornmeal broth +5% milk media/corn meal agar, colony morphology, sugar assimilation, and fermentation tests, morphology in maize agar, growth at 42 °C, and real-time PCR. Among these given the relatively higher sensitivity and specificity, the use of molecular tools like PCR should be preferred in the future studies. Further use of more than one diagnostic modality (combining one of the preliminary tests like CHROMagar with a molecular tool) could in turn increase the reliability of the results.

Although all included studies had well-defined cases (histopathological confirmed OSCC) and control (clinically confirmed normal oral mucosa), there were several liabilities due to the lack of adequate matching between the comparison groups. Among the seven studies, only four studies [36–38,40] accounted for known confounders (age and gender), while one [35] matched only for gender and the remaining two studies [34,39] either did not provide any matching information or failed to match the groups. Additional matching for potential confounders was made in only one [36] of the included studies wherein the denture status was considered. The lack of adequate matching resulted in a relatively lower score for the included studies on the Newcastle–Ottawa scale (Table 2).

In order to quantitatively assess the association between *Candida* prevalence, species specificity in OSCC, the odds ratio at 95% confidence interval was calculated for each of the included studies

(Table 3). Quantitative assessment was carried out only for those *Candida* species wherein the odds ratio was available from at least four studies. Forest plots were generated using the overall ratio (at 95% confidence interval) which in turn was calculated from the odds ratio obtained from the individual studies. Only total *Candida*, *CA*, and *NCA* association with OSCC had odds ratios from at least four studies and were assessed quantitatively. Based on the meta-analysis, a significant association was noted between both total *Candida* and OSCC. The result confirms the notion that *Candida* would be higher in the diseased state than the healthy controls. The clinical implication based on total *Candida* in OSCC is limited without knowing the species specificity. Similar to total *Candida*, even the *CA* showed a significant association with OSCC. Once again, *CA* being a commensal, its higher presence in OSCC just confirms its opportunistic nature. To establish a pathogenic aspect to *Candida* and OSCC, it was necessary to analyze the presence of the pathogenic *Candida*. Although *NCA* also showed an association with OSCC, the relationship was not significant. The nonsignificant association with *NCA* questions the notion that *NCA* could be a potential risk factor for OSCC. As mentioned earlier, since there were limited studies, it was not possible to analyze the association of OSCC with each *NCA* species individually. Thus, although overall *NCA* does not have a significant association with OSCC, it is unclear if there could be an *NCA* species-specific association with OSCC.

## 5. Conclusions

Based on the present systematic review and meta-analysis, OSCC has a significant association with *Candida*, and *CA*, while its association with *NCA* is not significant. A major parameter used to interpret the results of a qualitative and quantitative analysis would be the number and quality of the included studies. In the qualitative analysis, there were seven [34–40] studies included, with three [34,35,39] of them having scores below 7. In the quantitative analysis, there were only four to five studies included, wherein at least one of the included studies had a score less than seven. Thus, given the limited number of studies and their relatively lower Newcastle–Ottawa scores, the association noted in the present analysis must be inferred with caution. The main relevance of the present systematic review and meta-analysis was to assess the association of *Candida* and OSCC, and also to provide insight in to the quality and the quantity of studies estimating the same. As mentioned, there is a lack of both quality and quantity of studies analyzing the association between *Candida* and OSCC.

Future perspective: To improve the quality of the data, it is vital that future studies give sufficient importance to matching the comparison groups for both known and potential confounders. It is also of utmost importance that the studies are designed such that the examiners are blinded to the comparison groups.

**Funding:** This research received no external funding.

**Conflicts of Interest:** The authors declare no conflict of interest.

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
