# Peer review of "Analyzing the Association between Candida Prevalence, Species Specificity, and Oral Squamous Cell Carcinoma: A Systematic Review and Meta-Analysis— Candida and OSCC"

_applsci, doi:10.3390/app10031099_

Round 1
Reviewer 1 Report
Write Candida as italics throughout the manuscript. First line of the abstract is similar to title of the manuscript. Rewrite. Use abbreviated forms in the abstract as well eg.. OSCC, Candida..etc.There is a lack of searching time frame in the meta analysis. It is better to include time period used for the searching web data base. What is advantage of the present metaanalysis is not highlighted. Future perspective and advisory opinion of the authors is very significant contribution to the review. Add a note on it. References cited were not good enough for the manuscript.
Thus, such factors are termed potential risk factors. One such factor 50 for OSCC is micro-organisms – what are the other potential risk factors? What is the basis for inclusion and exclusion criteria? Newcastle Ottawa scale – include detailed procedure and significance of this method for meta-analysis. Also, provide or cite suitable reference for this method. The key words and focused question are not relevant to each other.
Focused question: “Is there an association between candida prevalence, species specificity, and OSCC? Keywords “candida and oral squamous cell carcinoma. What if any of the unrelated candida to OSCC problem.
Author Response
Comments and Suggestions for Authors
1) Write Candida as italics throughout the manuscript. First line of the abstract is similar to title of the manuscript. Rewrite. Use abbreviated forms in the abstract as well eg.. OSCC, Candida..etc.
Reply: As advised, candida is written in italics throughout the manuscript. As advised, the first line of the manuscript is changed. Instead of the full forms the abbreviated version including OSCC, C.albicans, non-C.albicanss are used in the abstract
2) There is a lack of searching time frame in the meta-analysis. It is better to include time period used for the searching web data base. What is advantage of the present meta-analysis is not highlighted. Future perspective and advisory opinion of the authors is very significant contribution to the review. Add a note on it. References cited were not good enough for the manuscript.
Reply: The search with the keywords was made on October 2019. It included all article published till that date. Thus, there was no time frame to mention. As advised, a sentence on the relevance of the present systematic review and meta-analysis is added to the conclusion of the review. The last few lines of the conclusion fell in the category of future perspective and as advisory notes; thus, they have been added under the subheading future perspective. As advised the references are increased to support the statements made in the review.
3) Thus, such factors are termed potential risk factors. One such factor 50 for OSCC is micro-organisms – what are the other potential risk factors? What is the basis for inclusion and exclusion criteria? Newcastle Ottawa scale – include detailed procedure and significance of this method for meta-analysis. Also, provide or cite suitable reference for this method.
Reply: As advised, the other potentially malignant disorders are enlisted with appropriate references. The reason for the specific inclusion and exclusion criteria used is mentioned immediately after the criteria’s under materials and methods. The scoring system for the Newcastle-Ottawa scale is already mention under materials and methods. The advised felt that since Newcastle-Ottawa scale is a relatively common system followed to assess the quality of non-randomized studies included in a systematic review and/or meta-analyses, it was not necessary to write up on the same.
4) The key words and focused question are not relevant to each other. Focussed question: “Is there an association between candida prevalence, species specificity, and OSCC? Keywords “candida and oral squamous cell carcinoma. What if any of the unrelated candida to OSCC problem.
Reply: The keywords “candida and oral squamous cell carcinoma’ is not specific. That was the intension of the author. The author wanted to make sure that the keywords provide a wide range of articles analysing both candida and OSCC irrespective of the nature of the study. The author felt that, the relatively non-specific keywords, would reduce the risk of missing any relevant articles.
Note: All the corrections are implemented in the revised manuscript in the form of track changes

Reviewer 2 Report
In the present review, the author evaluated the association between candida prevalence, species specificity, and oral squamous cell carcinoma through qualitative and quantitative analysis. Although the author performed a meta-analysis and claimed that oral squamous cell carcinoma has a significant association with candida and candida albicans, it lacks the scientific rigor and depth to support the claim and conclusion due to the limited number of studies (only 7). As a result, I don't recommend this review to be accepted in its current state.
Following are my questions and comments on the manuscript.
1. Keywords: “non-candida albicans” is not an eligible keyword. Please delete it.
2. The introduction should be intensively improved since it routinely introduces some background without real scientific depth. It is hard to understand the significance of this work from the presented introduction. In addition, many statements are too casual without any supporting references.
3. Lines 48-50: “Apart from these known risk factors, several factors have been implicated in the development of OSCC, although their evidence of association is limited.” Several factors? For example? Please provide the supporting references. As a scientific paper, such casual statement should be avoided unless solid evidence can be presented.
4. Line 51: “Studies have shown the presence of several oral microbes in OSCC patients [6-20]” Only 44 references were cited in this review, however, one-third of which (15 references) were listed here. It is unacceptable and the author should review properly the Author Guidelines of the Applied Sciences.
5. Line 58: “Among oral microbes, much importance has been given to human papillomavirus types 16 and 18, due to their proven association with cervical and oropharyngeal cancer” Any supporting references?
6. Line 108: “using the keywords “candida and oral squamous cell carcinoma” until October 2019.” The author claims that only 7 studies [22-28] satisfied the inclusion criteria and provided sufficient data for qualitative analysis. However, almost all of these 7 references published in 2018 and 2019. So I am wondering what is the start date of this research?
7. Lines 212-217: Now that so many additional diagnostic modalities were listed in the present review, why one is better? I recommend proceeding some comparative analysis rather than just presenting the methods.
Author Response
Comments and Suggestions for Authors
In the present review, the author evaluated the association between candida prevalence, species specificity, and oral squamous cell carcinoma through qualitative and quantitative analysis. Although the author performed a meta-analysis and claimed that oral squamous cell carcinoma has a significant association with candida and candida albicans, it lacks the scientific rigor and depth to support the claim and conclusion due to the limited number of studies (only 7). As a result, I don't recommend this review to be accepted in its current state.
Following are my questions and comments on the manuscript.
Keywords: “non-candida albicans” is not an eligible keyword. Please delete it.
Reply: As advised, the keyword is removed
The introduction should be intensively improved since it routinely introduces some background without real scientific depth. It is hard to understand the significance of this work from the presented introduction. In addition, many statements are too casual without any supporting references.
Reply: As advised, the introduction is revised by adding more information about the pathogenesis of a candida mediated carcinogenesis.
Lines 48-50: “Apart from these known risk factors, several factors have been implicated in the development of OSCC, although their evidence of association is limited.” Several factors? For example? Please provide the supporting references. As a scientific paper, such casual statement should be avoided unless solid evidence can be presented.
Reply: As advised, references are added to statement lacking the same.
Line 51: “Studies have shown the presence of several oral microbes in OSCC patients [6-20]” Only 44 references were cited in this review, however, one-third of which (15 references) were listed here. It is unacceptable and the author should review properly the Author Guidelines of the Applied Sciences.
Reply: As advised in query 3, more references were added for sentences lacking the same. Thus, now the overall references have increased. The references cited for microbes [6-20] have been retained as the authors felt the included references were appropriate for the statement made.
Line 58: “Among oral microbes, much importance has been given to human papillomavirus types 16 and 18, due to their proven association with cervical and oropharyngeal cancer” Any supporting references?
Reply: As advised, appropriate references have been added
Line 108: “using the keywords “candida and oral squamous cell carcinoma” until October 2019.” The author claims that only 7 studies [22-28] satisfied the inclusion criteria and provided sufficient data for qualitative analysis. However, almost all of these 7 references published in 2018 and 2019. So I am wondering what is the start date of this research?
Reply: The search with the keywords was made in October 2019. It included all article published till that date. Thus, there was no limitation or a starting date to mention. Also, when the studies were selected, only the inclusion and exclusion criteria were considered. It was a coincidence that all the article selected happened to be published in 2018, 2019. It was not intentional from the authors side.
Lines 212-217: Now that so many additional diagnostic modalities were listed in the present review, why one is better? I recommend proceeding some comparative analysis rather than just presenting the methods
Reply: As advised, the authors have included a few sentences emphasizing the need to use molecular diagnostic tools like PCR to increase the sensitivity and the specificity of the results. In addition, it is also stated that the use of more than one diagnostic modality could increase the reliability of the results.
Note: All the corrections are implemented in the revised manuscript in the form of track changes

Round 2
Reviewer 1 Report
the manuscript is good.
Author Response
Comments and Suggestions for Authors from Reviewer 1:
The manuscript is good
Reply: The author thanks the reviewer for the comments

Reviewer 2 Report
The authors claimed the all the corrections are implemented in the revised manuscript. Nevertheless, they did not really address these key questions raised by reviewers.
Here are some remarks to the revised manuscript:
1. The author states that the introduction has been revised by adding more information about the pathogenesis of a candida mediated carcinogenesis. In fact, the author only added some references. I don’t think the significance of this review could be clearly presented by doing so. The authors should explain how those results shape our current understanding of the topic and present a clear statement of what you intend to prove by citing these publications. In my view, this section should be polished.
2. The author affirms that the references cited for microbes [6-20] were appropriate for the statement made. After carefully reviewing all these cited references, I think Ref 15 should be deleted since it is a study about oral microbiota in pancreatic cancer rather than oral microbes in OSCC patients.
3. The author should double check all the references and ensure the style is consistent.
1) Page ranges. full or no? For example, ref 11 '304-8'; ref 12 '304-8'; ref 13 '711-719'; ref 15 '582–588'.
2) Article titles: Whether the initial letter of each significant word should be capitalized? Please check ref 18, 19, 27, 37, and 56.
Author Response
Comments and Suggestions for Authors from reviewer 2
The authors claimed the all the corrections are implemented in the revised manuscript. Nevertheless, they did not really address these key questions raised by reviewers.
Here are some remarks to the revised manuscript:
The author states that the introduction has been revised by adding more information about the pathogenesis of a candida mediated carcinogenesis. In fact, the author only added some references. I don’t think the significance of this review could be clearly presented by doing so. The authors should explain how those results shape our current understanding of the topic and present a clear statement of what you intend to prove by citing these publications. In my view, this section should be polished.Reply: Lines 73 to 86 were added in the last revision about the carcinogenicity of candida (highlighted in yellow). In the present revision, few lines are included to the end of the introduction explaining the purpose of the meta-analysis (highlighted in red) as follows:
The sole purpose of the meta-analysis was to estimate using the current published data if there is a significant difference in the candida prevalence and species specificity between oral cancer tissue and the normal oral mucosa. A significantly higher candida prevalence, and or a predominant NCA in OSCC compared to normal oral mucosa, would provide a preliminary evidence that candida, especially the treatment-resistant NCA have an affinity towards OSCC, although it would not be possible to comment if the higher prevalence and/or NCA specificity is a result of the OSCC or the OSCC is a result of the higher prevalence and/or NCA specificity.
As mentioned above, the aim of the manuscript was not to estimate if candida causes oral cancer. Thus, the authors have cited only a few references with respect to the carcinogenic potential of candida, as it is not the aim of the study.
As far as the results of the review and their impact of the current understanding of the topic is concerned, the results show a significant association, but the lack of quality studies in the meta-analysis has prevented the authors from making any conclusive remarks. The major output from the review is that there is a lack of quality studies comparing the candida prevalence and species specificity in oral cancer and normal oral mucosa. Thus, the authors have commented under the subheading future perspectives the following:
Future perspective: To improve the quality of the data, it is vital that future studies give sufficient importance to matching the comparison groups for both known and potential confounders. It is also of utmost importance that the studies are designed such that the examiners are blinded to the comparison groups.
In addition, a few lines were also included in the discussion in the previous revision as follows:
Among these given the relatively higher sensitivity and specificity, the use of molecular tools like PCR should be preferred in the future studies. Further use of more than one diagnostic modality (combining one of the preliminary tests like CHROMagar with a molecular tool) could in turn increase the reliability of the results.
The author affirms that the references cited for microbes [6-20] were appropriate for the statement made. After carefully reviewing all these cited references, I think Ref 15 should be deleted since it is a study about oral microbiota in pancreatic cancer rather than oral microbes in OSCC patients.
Reply: As advised reference 15 is replaced by the following appropriate reference:
Zhang, Z.; Yang, J.; Feng, Q.; Chen, B.; Li, M.; Liang, C.; Li, M.; Li, Z.; Xu, Q.; Zhang, L.; Chen, W. Compositional and Functional Analysis of the Microbiome in Tissue and Saliva of Oral Squamous Cell Carcinoma. Front Microbiol. 2019, 10, 1439.
The author should double check all the references and ensure the style is consistent. a) Page ranges. full or no? For example, ref 11 '304-8'; ref 12 '304-8'; ref 13 '711-719'; ref 15 '582–588'.
Reply: As advised, the page number range of all the references are made uniformly full
b) Article titles: Whether the initial letter of each significant word should be capitalized? Please check ref 18, 19, 27, 37, and 56.
Reply: As advised, the initial letters are made uniformly small for references 18,19,27,37, and 56.
Note: The changes made in the revised draft (2nd revision) are highlighted in red

Round 3
Reviewer 2 Report
The author have addressed my concerns in the revision.
Author Response
Comments and Suggestions for Authors from reviewer 2:
The author has addressed my concerns in the revision:
Reply: The authors thank the reviewer for the comments
